# Identification of the onchocerciasis vector in the Kakoi-Koda focus of the Democratic Republic of Congo

**Rory J. Post[1,2]☯, Anne Laudisoit[3,4,5]☯, Michel Mandro[6], Thomson Lakwo[7], Christine Laemmer[8,9], Kenneth Pfarr[8,9], Achim Hoerauf[8,9], Pablo Tortosa[10], Yann Gomard[10], Tony Ukety[11], Claude Mande[12,13], Lorne Farovitch[14], Uche Amazigo[15], Didier Bakajika[16], David W. Oguttu[7], Naomi Awaca[17], Robert Colebunders[4]\***

1 Disease Control Department, London School of Hygiene & Tropical Medicine, London, United Kingdom, 2 School of Biological and Environmental Sciences, Liverpool John Moores University, Liverpool, United Kingdom, 3 EcoHealth Alliance, New York, New York, United States, 4 Global Health Institute, University of Antwerp, Wilrijk, Belgium, 5 Evolutionary Ecology group (EVECO), University of Antwerp, Wilrijk, Belgium, 6 Provincial Health Division Ituri, Ministry of Health, Bunia, Democratic Republic of Congo, 7 Division of Vector Borne and Neglected Tropical Diseases Control, Ministry of Health, Kampala, Uganda, 8 Institute for Medical Microbiology, Immunology and Parasitology, University Hospital Bonn, Bonn, 9 German Center for Infection Research (DZIF), partner site Bonn-Cologne, Germany, 10 Université de La Réunion, UMR PIMIT "Processus Infectieux en Milieu Insulaire Tropical", CNRS 9192, INSERM U 1187, IRD 249. Plateforme de Recherche CYROI, Saint-Denis, France, 11 Centre de Recherche en Maladies Tropicales (CRMT), Rethy, Democratic Republic of Congo, 12 Department of Ecology and Wildlife Management, University of Kisangani, Kisangani, Democratic Republic of Congo, 13 Biodiversity Monitoring Centre (CSB), University of Kisangani, Kisangani, Democratic Republic of Congo, 14 School of Medicine and Dentistry, University of Rochester Medical Center, Rochester New York, United States of America, 15 Pan-African Community Initiative on Education and Health, Enugu, Nigeria, 16 Expanded Special Project for Elimination of NTDs, World Health Organization Regional Office for Africa, Brazzaville, Republic of Congo, 17 Ministry of Health, National Programme for Neglected Tropical Diseases & Preventive Chemotherapy, Kinshasa, Democratic Republic of Congo

☯ These authors contributed equally to this work.

\* robert.colebunders@uantwerpen.be

**Data Availability Statement:** All relevant data are within the manuscript and its Supporting Information files.

## Abstract

### Background

The objective of this study was to characterise the vector in a small hyper-endemic focus of onchocerciasis (the Kakoi-Koda focus) which has recently been discovered on the western slopes of the rift valley above Lake Albert.

### Methodology/Principal findings

Aquatic stages of blackflies were collected by hand from streams and rivers, and anthropophilic adult females were collected by human landing catches. Using a combination of morphotaxonomy and DNA barcoding, the blackflies collected biting humans within the focus were identified as *Simulium dentulosum* and *Simulium vorax*, which were also found breeding in local streams and rivers. *Simulium damnosum* s.l., *Simulium neavei* and *Simulium albivirgulatum* were not found (except for a single site in 2009 where crabs were carrying *S. neavei*). Anthropophilic specimens from the focus were screened for *Onchocerca* DNA using discriminant qualitative real-time triplex PCR. One specimen of *S. vorax* was positive

**Funding:** This work was supported by the European Research Council (ERC 671055 to RC) and VLIRUOS (2015-2017, project "Reducing the Incidence of River Epilepsy in Orientale Province DRCongo" to RC), and the Belgian Technical Cooperation (CTB; RDC1015311/PEE/12/2014 Fonds de l 'expertise, 2016, project "EPIRIVE" to AL and RC), and the National Science Foundation under Grant No. DEB-1618919 to EcoHealth Alliance (EcoHealth Net 2.0; Lorne Farocitch). The funders had no role in study design, data collection and analysis, decision to publish, or preparation of the manuscript.

**Competing interests:** No authors have competing interests.

for *Onchocerca volvulus* in the body, and out of 155 *S. dentulosum*, 30% and 11% were infected and infective (respectively).

## Conclusions/Significance

*Simulium dentulosum* currently appears to be the main vector of human onchocerciasis within the Kakoi-Koda focus, and *S. vorax* may be a secondary vector. It remains possible that *S. neavei* was the main (or only) vector in the past having now become rare as a result of the removal of tree-cover and land-use changes. *Simulium vorax* has previously been shown to support the development of *O. volvulus* in the laboratory, but this is the first time that *S. dentulosum* has been implicated as a probable vector of onchocerciasis, and this raises the possibility that other blackfly species which are not generally considered to be anthropophilic vectors might become vectors under suitable conditions. Because *S. dentulosum* is not a vector in endemic areas surrounding the Kakoi-Koda focus, it is probable that the Kakoi-Koda focus is significantly isolated.

### Author summary

River blindness (= onchocerciasis) is a severely debilitating disease caused by the nematode parasite *Onchocerca volvulus*, and in Africa it is known to be transmitted from person to person by blood-sucking blackflies (Diptera: Simuliidae) of the *Simulium damnosum* complex, the *S. neavei* group and (rarely) by *S. albivirgulatum*. Using classical morphological characteristics and DNA analysis we have unexpectedly identified the vector blackfly as *S. dentulosum* (and possibly *S. vorax*) in a small endemic area (the Kakoi-Koda focus) which has been recently discovered on the western slopes of the rift valley in the Democratic Republic of Congo above Lake Albert. In the surrounding endemic areas, the vectors are *S. damnosum* complex and/or *S. neavei* (as normally expected), and because *S. dentulosum* is not a vector in these surrounding areas, it follows that this focus is entomologically isolated from immigrant blackfly species which might otherwise have carried new infections into the Kakoi-Koda focus (and vice-versa). This is of local importance, because it makes elimination of the parasite easier, but our findings have wider significance across Africa, because they raise the possibility that under the right conditions, other common and widely-distributed blackfly species might unexpectedly become human-biters and significant vectors.

## Introduction

### Historical introduction

*Onchocerca volvulus* was initially recorded within the borders of the modern-day Democratic Republic of Congo (DRC) by Émile Brumpt in 1903, only ten years after the parasite had been described and named [1]. During the next 50 years, the amount of work carried out on onchocerciasis in DRC was prolific and far exceeded that occurring in any other African country [2]. However, coverage was uneven in the remoter parts, and as a consequence there remained gaps within the comprehensive mapping of endemic areas and vectors, including the highland parts of modern-day Ituri Province (especially Mahagi, Djugu and Aru Territories).

In their monograph on the distribution of onchocerciasis in DRC, Fain and Hallot [3] recorded onchocerciasis from the downstream environs of the Ituri River (which they called the "*foyer de la rivière Aruwimi et de ses affluents—Nepoko et Ituri*"). However, they did not record it from the Ituri Highlands, which form the western edge of the Albertine Rift along the shores of Lake Albert, and is where the Ituri River has its source. The sites where onchocerciasis was known within the River Aruwimi focus are approximately 250 km west of the Ituri Highlands, except for two (at Ofay and Vieux-Kilo) which are just at the southern edge, but at least 90 km southwest from the Kakoi-Koda focus. Fain and Hallot [3] stated that both onchocerciasis and *S. neavei* were absent from "Haut-Ituri", and whilst *S. damnosum* s.l. was present, it was not anthropophilic. It is unclear exactly where they meant by "Haut-Ituri", but the most common usage is that defined by Misonne [4], (who was working more or less at the same time as Fain and Hallot [3], as "a high plateau in the territories of Djugu and Mahagi". This would place it in the northern part of the Ituri Highlands (which are more or less identical to the area defined by Wiese, 1979, as the Blue Mountains), and would be almost contiguous with the Kakoi-Koda focus situated on the eastern slopes of the Blue Mountains which form a scarp face and drop steeply towards Lake Albert in the bottom of the rift valley (Fig 1).

Raybould and White [5] interpreted Fain and Hallot [3] to indicate a downstream focus (centred on the Ituri River and its tributaries) which they renamed the 'Ituri Focus'. Fain and Hallot's 1965 map was later updated and republished by both Maertens in 1990 [1] and Fain in 1991 [6], but there were no additional data for the Ituri Highlands (including the scarp face). However, in 1995 WHO [7] indicated on their map that the whole of Ituri Province, including the Ituri Highlands, was endemic. It is unclear how WHO came to this conclusion because no data sources were cited, but they also stated that "in . . . Zaire, the current situation is uncertain", and it is therefore likely that in Ituri their map was the result of interpolation of the known endemic areas in an uncertain situation. Whatever the explanation, in the same year (1995) WHO also established the African Programme for Onchocerciasis Control (APOC) with the objective of controlling onchocerciasis so that it was no longer a public health problem in 20 remaining endemic countries in Africa outside the former Onchocerciasis Control Programme (OCP). As part of this, DRC carried out Rapid Epidemiological Mapping of Onchocerciasis (REMO) throughout the whole country between 1997 and 2008 with the objective of defining hyper- and meso-endemic areas where onchocerciasis would be controlled by Community Directed Treatment with Ivermectin (CDTI) [8]. Endemicity in areas with less than 20% prevalence of nodules was considered to be sporadic or hypo-endemic and to be of no public health importance. Results from the REMO for Ituri Province indicated that hyperendemic onchocerciasis was more or less continuously distributed throughout most of the province, forming a single very large transmission area (the North-Eastern DRC Oncho Endemic Area) which also crossed over the local borders into Nord Kivu, Tshopo and Haut Uélé Provinces, and extended further into Bas Uélé, Tshuapa and Maniema Provinces. The exception to this pattern was in the Ituri Highlands, where onchocerciasis was mostly hypoendemic except for a small area of hyper-endemism in the northeast corner above Lake Albert (with 40% or higher nodule prevalence—see Fig 2 published by Zouré et al. [9], and Fig 1 published by Makenga Bof et al. [8]. We propose to call this small outcrop of hyper-endemism the Kakoi-Koda focus (because the focus is more or less bounded by these two rivers–Fig 2A). Our study was centred upon this focus which had already been found to have a high prevalence of epilepsy [10] and has been subject to further investigations into the link between epilepsy and onchocerciasis [11, 12]. The entomological work carried out within the Kakoi-Koda focus was originally proposed to provide the entomological component for those investigations (2015–2017), and the surveys of the surrounding areas in 2009 and 2016 were originally carried out as

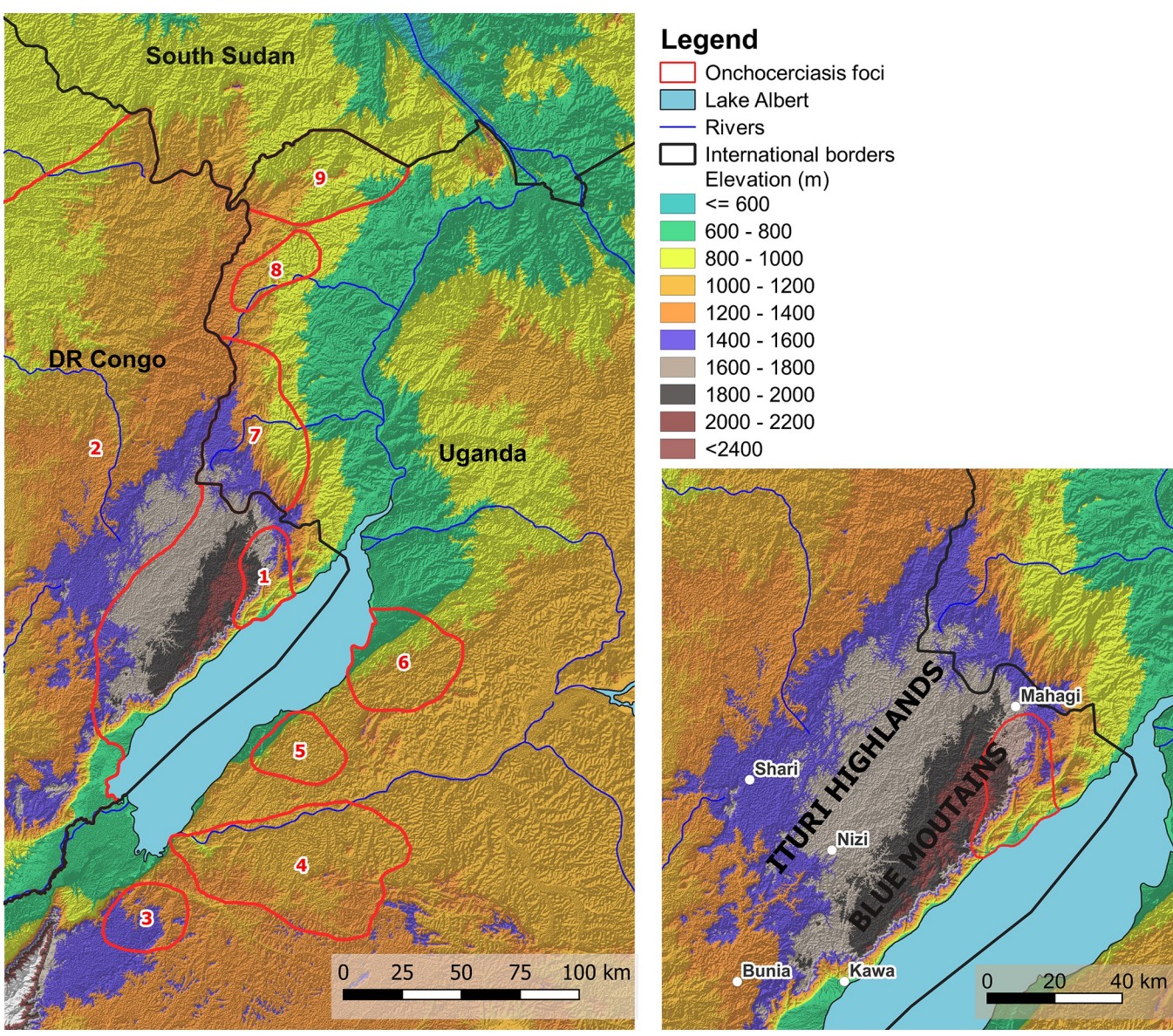

**Fig 1. Relief map showing location of the Kakoi-Koda focus and other nearby foci. 1** = Kakoi-Koda focus; **2** = North-Eastern DRC Oncho Endemic Area; **3** = Itwara focus; **4** = Mpamba-Nkusi focus; **5** = Wambabya-Rwamarongo focus; **6** = Budongo focus; **7** = Nyagak-Bondo focus; **8** = Maracha-Terego focus; **9** = West Nile focus. Location of onchocerciasis foci according to Katabarwa et al. [29] and Makenga Bof et al. [8]. (Produced in QGIS: [50]). The RASTER images used to produce Fig 1 are Landsat-7 and Landsat-8 images, courtesy of the U.S. Geological Survey (https://www.usgs.gov/). The basic shapefiles for administrative areas can be found on the Référentiel Géographique Commun for the DRC (www.rgc.cd).

part of a DRC-Uganda collaboration to characterise the potential for cross-border reinvasion of onchocerciasis foci prior to proposed elimination activities.

## Ecological description of the study area

The Kakoi-Koda focus of onchocerciasis occupies a small area in the north east of the Ituri Highlands on the scarp slope which drops steeply down from the Blue Mountains near Mount Aboro towards Lake Albert (Figs 1, 2, S1 and S2). The area of the focus is approximately 715 km$^2$ and it has a perimeter of more or less 115 km. It lies mostly between the Rivers Kakoi and

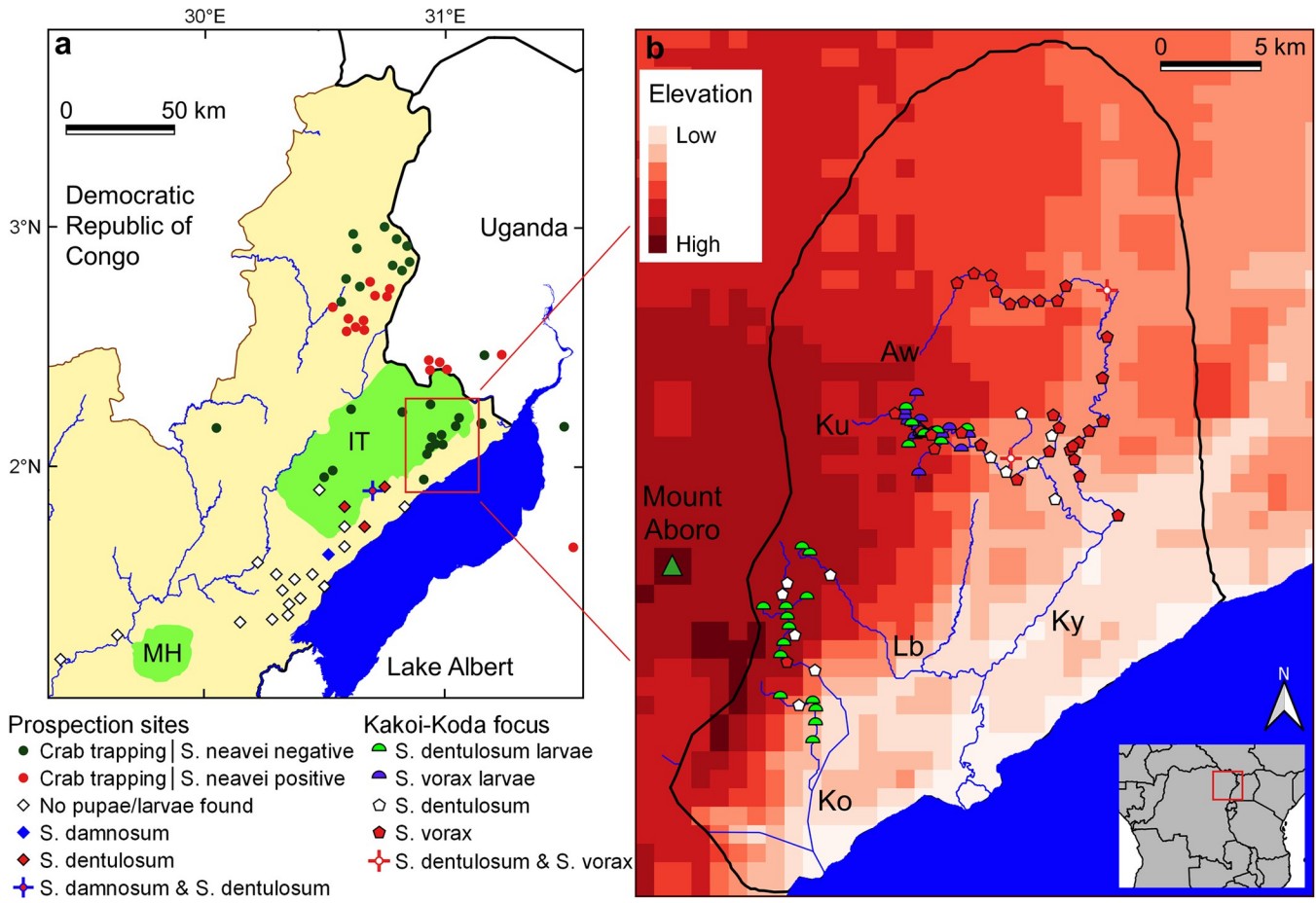

**Fig 2. Overview map of historical and recent surveys for *Simulium* spp. A.** Prospection sites in the Ituri highlands (IT), around the Mount Hoyo Reserve (MH) and some main rivers. **B.** Kakoi-Koda focus with sampled streams (Aw: Awoo, Ku: Kuda, Lb: Lebu, Ky: Kakoi, Ko: Koda) (produced in QGIS [50]).

Koda, with the Rivers Kuda and Lebu running through it. These rivers flow in a generally easterly to southerly direction off the Ituri Highlands plateau, down the scarp face and into Lake Albert (Figs 2B, S1 and S2).

The Ituri Highlands form the western edge of the rift valley. They are the result of geological uplift and subsequent fracture of the continental plate, such that the fracture line forms the rift valley (the Albertine Rift) as the two halves of the uplifted and broken plate are slowly drawing apart. The Ituri Highlands forming the western side of the rift valley consist of a 'tilted plateau', with a dip slope which rises gently from west to east, and then falling sharply down a scarp slope into the rift valley by Lake Albert [13]. The altitude of the dip slope varies between 1,700 m and 2,000 m, rising towards its eastern edge which is crested by a series of cone-shaped mountains, the highest of which is Mount Aboro (2,455 m). The tilted plateau which forms the Ituri Highlands (including the scarp face) is formed of hard Precambrian rocks, but there is a band of Holocene sediments at the foot of the scarp slope along the shores of Lake Albert [13].

Whilst the geological structure is clear, it is unfortunate that geographical descriptions of the area are sometimes confusing because the various geographic names have been used inconsistently and in contradictory ways by different authors and cartographers. Wiese [13] and some cartographers have called this whole area the Blue Mountains (except for the scarp slope), but most authors and the local residents restrict the term 'Blue Mountains' to the line

of cone-shaped mountains along the crest of the tilted plateau where the dip slope meets the scarp slope. Wiese [13] also excludes most of the scarp slope from his definition of the Blue Mountains, but they were given this name by Samuel Baker in 1866 and it is clear that Baker intended the name to apply to the whole of the scarp slope from where it rises up from the bottom of the rift valley, near Lake Albert, to the highest peaks [14]. The term 'Ituri Highlands' is more or less identical to the area defined by Wiese [13] as the 'Blue Mountains' (except that the 'Ituri Highlands' includes the scarp slope), and the area described by Fain and Hallot [3] as 'Haut Ituri' would include only the northern part of the Ituri Highlands.

The vegetation cover on the Ituri Highlands is mostly savanna (of various sorts) except for three main areas of forest. Meessen [15] and Weise [13] are consistent with each other in their descriptions of there being Equatorial Rain Forest mostly distributed to the West of the Ituri Highlands, but extending eastwards in a mosaic towards the Shari and Nizi Rivers. Secondly, they also describe Montane Forest found above 2,000 m on the highest peaks above the scarp slope. The plateau is described as being mainly 'Grass Savanna' with relatively few trees (except *Erythrina* sp). Weise [13] described the montane forests as 'relicts', speculating that they had once been continuous with the Congo Basin Equatorial Rain Forest, and that they had been separated by anthropogenic deforestation. However, modern authors have questioned this narrative [16]. It seems that the area would have been deforested 18,000 years ago as part of the world-wide retreat of the tropical rain forests during the last of the Pleistocene glaciations, but by the Holocene 'climatic optimum' (which was warm and damp, 8,000 years ago) it had probably become reforested, only to become deforested again after the aridification event that marked the beginning of the Meghalayan Age (approx. 4,200 years BP) when the climate took a sharp turn towards cold and dry [17]. By this time modern humans (*Homo sapiens*) were long-established and it is not clear the extent to which anthropogenic clearances might also have contributed. Whatever the ancient ecological history of the Ituri Highlands, aerial photographs from 1953 (Royal Museum for Central Africa, Brussels) show that the general pattern of the current-day distribution of forest and savanna also existed at least 70 years ago (as described by Meessen [15]). It is also probable that this basic pattern was already in existence more than 110 years ago because the first useful map of the area on both sides of the Rift Valley was published by Bright [18] and he indicated as "dense forest" those areas which are still considered to be of a forest biotype, but the plateau was not indicated as such. Furthermore, he described that "the country in the mountains is a charming one; the grass is generally short, and it is easy to travel from one place to another" and "looking to the north-eastwards from Abura [= Aboro] . . .. the country appears to be hilly and wooded", and a few published photographs of the plateau and the scarp show a savanna biotype [18, 19]. This does not mean that there has not been significant and continuing anthropogenic deforestation and bush clearance of the montane forests and the savanna woodlands respectively, because this was directly observed and reported by Wiese [13], and whilst the basic historical pattern can still be seen on modern satellite imagery (Google Earth, accessed 08.xii.2019), it is clearly more degraded. Indeed, from 2001 to 2018, the tree cover loss amounted to 4.5% and 5.2% respectively in the Mahagi and Djugu territories (across which the Kakoi-Koda focus lies) [20]. The Mount Aboro nature reserve has become totally deforested [21].

The vegetation of the scarp slope facing Lake Albert has been described as 'Dry Savanna' near the bottom of the slope, with *Acacia* sp., *Albizzia* sp., *Euphorbia* sp. and *Kigelia africana*, and further up the slope where there is more rainfall, the vegetation is generally 'Park Savanna' with light tree cover and *Digitaria* short grass cover in the higher parts [13, 15, 22]. However, it is on the scarp slope that a third area of closed-canopy forest has recently been described by Laudisoit et al. [21]. Situated below Mount Aboro, it partly overlaps the Kakoi-Koda focus, and covers a wide altitudinal range upwards from near the lake shore. This forest area is

currently fragmented (S2 Fig) but was probably much more extensive in the past, with current rates of deforestation estimated at 1.2% per year [21]. The forest fragments consist predominantly of secondary forest due to the extensive human activities while the larger montane forest galleries consist of a mixture of secondary and near-primary forests dominated by Myristicaceae, Sapotaceae, Annonaceae, Apocynaceae and Euphorbiaceae.

It can rain in any month of the year in the environs of the Kakoi-Koda focus, but during the driest months (December to March) there is only 50 mm or less rainfall per month. The rainy season is weakly bimodal with peaks of rainfall in May (with nearly 150 mm) and August (with nearly 200 mm) [13].

The Kakoi and Koda Rivers, associated with the focus, have their sources along the Blue Mountains watershed which separates water draining eastwards into Lake Albert (in the rift valley) and thence into the River Nile, from water draining westward into the Congo basin. The sources of these rivers are both east and west of Mount Aboro, but their upper courses are mostly in broad, flat, marshy valleys on the plateau, which then discharge down the scarp face where their courses are steep and cascading, before finally disgorging onto the flat sedimentary plain which forms the beach for Lake Albert in the Rift Valley [15]. The vegetation cover and land-use around the Kulubu Valley (which is drained by the River Kuda in the heart of the Kakoi-Koda focus) consists of a mosaic *Erythrina*-dominated dry savanna, rock outcrops, agricultural lands and relict montane forest galleries. Rice fields, palm nut plantations and banana trees are found on the lower slopes along the watercourses that flow into Lake Albert. The land around the steeper middle section is less densely populated than the lakeside or the top of the escarpment, but it is in these villages that onchocerciasis is hyperendemic and the focus is found.

## Potential isolation of the focus

DRC established 22 CDTI (Community Directed Treatment with Ivermectin) projects against onchocerciasis in 2012, including two projects in Ituri Province (Ituri North and Ituri South) [23], and our study area was within the Ituri North CDTI project. The initial objective of these CDTI projects in DRC (along with all other APOC countries) was to control onchocerciasis, and hence most of the Ituri Highlands (specifically Aru and Logo Health Zones) were excluded from CDTI because it was hypo-endemic or sporadic (i.e. less than 20% nodule prevalence by REMO). However, the recent change of objective from onchocerciasis control to elimination of transmission requires new epidemiological mapping (known as 'elimination mapping', using OV-16 serology) so that 'Transmission Zones' (TZs) can be properly defined and CDTI can be extended into hypo-endemic areas [24–26]. A TZ is defined as "*a geographical area where transmission of* Onchocerca volvulus *occurs by locally breeding vectors and which can be regarded as a natural ecological and epidemiological unit for interventions*" [26]. This implies that a TZ is sufficiently isolated from immigration of parasites (carried by infected vectors or humans) such that progress towards elimination in one TZ will be unaffected by what is happening in neighbouring TZs. There will normally be an onchocerciasis-free buffer zone in-between adjacent TZs. The flight range and ability of vectors to migrate between TZs is influenced by the topology of the landscape and the identity of the vector. For example, *Simulium neavei* seems to show very little migration [27], and hence Garms et al. [28] were able to apply larvicides to eliminate adjacent populations of *S. neavei* one by one from neighbouring river systems in the Itwara focus of onchocerciasis in Uganda. Similarly, Budongo, Wambabya and Mpamba Nkusi foci on the eastern escarpment of the Albertine rift valley in Uganda appear to be close to each other (Fig 1), but were independently isolated and quickly achieved elimination because of the short flight range of *S. neavei* [29, 30]. By contrast, *S. damnosum* s.str. in

Uganda has been known to disperse and carry parasites up to 50 km from its breeding sites on the Victoria Nile before it was eliminated [30], and some species in West Africa can migrate with the prevailing winds up to 500 km [31]. It is therefore important for the development of an appropriate onchocerciasis elimination strategy to identify the vector in the Kakoi-Koda focus, and to assess whether the focus is independent of surrounding areas and, in effect, is a TZ in its own right.

## Vector background

Worldwide, all known vectors of human onchocerciasis belong to the family Simuliidae (Diptera), and the only vectors known from Africa are members of either the *Simulium damnosum* complex (subgenus *Edwardsellum*) or the *Simulium neavei* group (subgenus *Lewisellum*) [32], except in parts of central DRC where *Simulium albivirgulatum* (subgenus *Metomphalus*) is the vector [33]. The only historical records of Simuliidae from Ituri Province in DRC were published by Fain in 1951 [34], who did not find any of these known vectors in the Ituri Highlands. However, he did not explore the northern part of the Ituri Highlands (which includes the area of the current-day Kakoi-Koda focus), and he did find *S. damnosum* s.l. breeding in the Ituri Highlands a little more than 30 km southwest of the Kakoi-Koda focus, but these were almost certainly non-anthropophilic (non-vector) cytospecies, because he specifically stated that they were not found to be biting humans there [3]. The nearest biting record for human-biting *S. damnosum* s.l. was from Camp Putnam near where the Kisangani-Bunia road crosses the River Epulu (approximately 160 km west of the Ituri Highlands). The nearest record of *S. neavei* in DRC was a single specimen collected biting man at Mount Hoyo, a little more than 50 km southwest of the Ituri Highlands [3]. However, *S. neavei* has long been known to be the vector of onchocerciasis across the border in Uganda, in the Nyagak-Bondo focus situated in current-day Nebbi, Zombo and Arua Districts (Fig 1). Nelson [35] showed that onchocerciasis was at high prevalence in Arua District in all communities from the top to the bottom of the scarp face (which forms the western edge of the Rift Valley, in a way that is similar to the situation of the Kakoi-Koda focus in DRC), and *S. neavei* was breeding in the streams and rivers flowing down the scarp, although adult flies were rare at most seasons [36]. Before any sort of control measures, the infection level (measured by prevalence rates of microfilariae in skin snips) in Nebbi District (at Agweci, on the River Nyarwodho close to the DRC-Uganda border, and just 10 km from the Kakoi-Koda focus), was 96%, and a mean of 95% across five communities in Zombo District [37]. The vector in the Nyagak-Bondo focus was *S. neavei* [29], but the Nkusi form of *S. damnosum* s.l. has also been reported from the Nyagak river in Uganda [38] and a small number of *S. vorax* were collected in November 2011 near the same river during human landing catches reported by Katabarwa et al. [37]. The source of this river is very close to the source of the Kakoi River on the Ituri Highlands. However, the Nkusi form of *S. damnosum* s.l. does not bite humans in Uganda and hence cannot be considered a vector [39], and the anthropophilic *S. vorax* was present in insignificant numbers. Transmission of onchocerciasis seems to have been interrupted in the Nyagak-Bondo focus since 2014 as a result of CDTI and vector elimination [29].

*Simulium neavei* is a simuliid species belonging to the subgenus *Lewisellum* and until recently it was thought to be the only vector species within *Lewisellum* from DRC [7, 34]. However, Makenga Bof et al. [23] have recently reported the presence of *S.* (*Lewisellum*) *woodi* in Ituri on the basis of information contained within the national strategic plan for the control of NTDs in DRC using preventative chemotherapy [40]. The source of this surprising but important claim is unclear. It is generally thought that *S. woodi* is restricted to Malawi and Tanzania [41, 42], and it has been implicated as a vector of onchocerciasis in both countries

[5]. However, Freeman and de Meillon [43] and Fain and Hallot [3] also list old records of *S. woodi* caught biting humans in Zambia, near the border with DRC.

### Animal *Onchocerca* species

The only records of animal-infecting *Onchocerca* species in DRC from near the Kakoi-Koda focus have been published by Rhodain and Gillain [44, 45] from two wild Buffalos (*Syncerus caffer*) near Nioka and near Aru, and they suggest that *Onchocerca* infested domestic goats were brought from the south to Aru. The Buffalos are now thought to have been infected by *Onchocerca dukei* [46], a species which is a parasite of domestic cattle and can be transmitted by *S. vorax* [47] and possibly *S. damnosum s.l.* [48]. *Onchocerca ochengi* has not been recorded from the study area, but it is a very widespread African parasite of cattle that is transmitted by *S. damnosum* s.l., and it is known from *S. damnosum* s.l. in Northern Uganda [49].

### Objectives of the study

The objectives of this study were to identify the vector of onchocerciasis in the Kakoi-Koda focus, and to try to assess whether the focus might be isolated from surrounding onchocerciasis-endemic areas in Ituri Province (DRC) and in northern Uganda. This is expected to be useful for planning, monitoring and understanding progress towards onchocerciasis elimination in Ituri and adjacent areas of Uganda. The study involved prospection of vector breeding sites and collection of anthropophilic blackflies both within and around the Kakoi-Koda focus, and the screening of the anthropophilic species for infection by *O. volvulus*.

## Methods

### Ethics statement

The study was approved by the Ethics Committee of the University of Antwerp B300201525249 and the Ethics Committee of Ngaliema Hospital, Kinshasa, Democratic Republic of Congo P: Eth/436/2015. Blackfly collectors were adults over the age of 18 years from local communities who provided written informed consent. They were given ivermectin before participating in blackfly collections.

### Field collections and other sources of material

Vector surveys were carried out in October 2009 in Mahagi Territory and Nebbi District (DRC and Uganda respectively) and again in Mahagi Territory in April-May 2016, in both years as part of a cross-border collaboration between the two countries [51]. Rivers were prospected for larval stages of *S. damnosum* s.l. and *S. neavei*, and vector collections were made by Human Landing Catches (HLCs) for short periods of time (30–60 minutes duration–sometimes known as 'spot checks') at each prospection site. Specimens were identified and counted, but not subject to further examination. During 2011, human landing catches were carried out from near the Nyagak river in the Nyagak-Bondo focus in Uganda. The methods and findings with respect to *S. neavei* have already been published [37], but here we report the numbers of other blackfly species which were caught in November 2011.

Collections were also made specifically from within the Kakoi-Koda focus and nearby sites expanding over the Djugu and Mahagi Territories in September-October 2015, August 2016, April to November 2017 and February to April 2018. Anthropophilic adult blackflies were collected using informal HLCs (*i.e.* more or less according to the methods described by Davies & Crosskey [52], but without prior knowledge of vector breeding sites to identify first-line villages, without rotation of vector collectors and without a formal daily timetable for collections

and vector collectors were offered ivermectin). Specimens were preserved in 70% ethanol. Streams and rivers were prospected for larvae and pupae of any species of Simuliidae, which were removed by hand from underwater substrates such as vegetation and rocks according to Davies and Crosskey [52]. Particular efforts were made to prospect sites which looked suitable for *S. damnosum s.l.* (*i.e.* white-water rapids), but prospections were not limited to these sorts of sites.

In 2009 and 2016 crabs were trapped using baited basket traps to determine the infestation rate of *S. neavei*. The traps, baited with meat, were placed in suitable places in deep water and left for one hour before retrieval and examination according to Garms et al. [28]. In 2015 and 2016, another type of locally made basket trap was used within the Kakoi-Koda focus and baited with fish or cassava, placed in the fast-flowing sections of the rivers surveyed. Because crab collections were poor, it seemed that crabs did not easily enter the locally-made basket, and crabs were subsequently hand-trapped. Crabs were sought in slow moving sections of the river, under stones and vegetation in a 100m transect of the stream or river for two hours by two people. Crabs which were caught by trap or by hand were checked for the presence of immature stages of *S. neavei* according to Davies & Crosskey [52]. Within the Kakoi-Koda focus, rivers were selected for survey according to their proximity to communities with *Onchocerca*-infected persons.

Larvae and pupae of all Simuliidae were preserved in spirit (ethanol or Carnoy's fixative), except some pupae which were maintained *en-masse* at ambient temperatures in humid vials in the laboratory for a few days to collect any neonate adult blackflies, which were then preserved in 70% ethanol.

Museum specimens (including adult males, adult females, pupae and larvae) of *S. dentulosum* and other species belonging to the subgenus *Anasolen* (including nomenclatural 'type-specimens') were examined from the collections of the Natural History Museum in London (United Kingdom) and the Royal Museum for Central Africa in Tervuren (Belgium).

## Taxonomic identifications of blackflies and morphotaxonomy of *S. dentulosum*

Larvae, pupae and adult blackflies were identified according to Freeman and de Meillon [43], Crosskey [53, 54] and Davies and Crosskey [52]. However, preliminary examinations of specimens collected from the Kakoi-Koda focus showed that neither the adult females nor the pupae of specimens similar to *S. dentulosum* could be easily identified using the usual taxonomic works, and further to this, specimens were collected biting humans (which is unusual for *S. dentulosum*). These specimens clearly belonged to the subgenus *Anasolen* according to Crosskey [54], but pupae had 15 gill filaments (instead of 14, which is expected in *S. dentulosum*), and adult females had a pleural membrane without hairs. In view of this unique combination of characters and the unusual anthropophily, a detailed morphotaxonomic study of the specimens was undertaken to assess their taxonomic status.

The morphology (structure, shape and colour) of *dentulosum*-like specimens from the Kakoi-Koda focus was compared with: 1) each other, and 2) with other *Anasolen* specimens (including *S. dentulosum*) from other localities, and 3) with the morphotaxonomic descriptions published by Fain [34, 55], Freeman and de Meillon [43], Grenier and Ovazza [56], Crosskey [53, 54], Grenier et al. [57], Fain et al. [58], Gouteux [59] and de Moor [60].

All specimens that were examined are listed below, and are all attributed to *S. dentulosum* unless otherwise stated. Where known, N/E co-ordinates are expressed in decimal degrees; dates of collection are day.month.year; L/LL indicates the number of larvae examined and

likewise P/PP for pupae, M/MM for adult males which were all reared from pupae (neonates), and F/FF for adult females which were also reared from pupae unless indicated HLC which were appetitive females caught in human landing catches.

Recent specimens from the environs of the Kakoi-Koda onchocerciasis focus in DRC included: **R. Koda at Rassia** N02.00021/E30.90450, 21.iv.17, 5 LL, 4 PP; **R. Lodda at Zaambi** N02.04862/E30.91464, 19.iv.17, 7 LL; **R. Koda at Adrasi** N02.12322/E30.96900, 24.ix.15, 24 LL, 3 PP; **R. Koda at Adrasi** N02.12322/E30.96900, 20.x.15, 7 PP; **R. Koda at Ndroi** N02.10425/E30.97488, 19.x.15, 3 LL, 1 P, 1 M (neonate); **R. Koda at Djupacora** N02.11101/E30.96219, 26.x.15, 1 L; **R. Koda at Gono** N02.01104/E30.90872, 23.x.15, 3 PP; **R. Madai at Bala** N02.02959/E30.90721, 26.ix.15, 5 PP; **R. Gridda at Ndeke** N01.98321/E30.91695, 24.x.15, 3 PP, 2 MM (neonate); **R. Lodda at Zaambi** N02.04862/E30.91464, Oct 2015, 2 PP; **R. Koda at Bala** N02.02959/E30.90721, Oct 2015,12 LL; Awoo waterfall at **Djupudnik** N02.16494/E30.99906, 22.viii.17, 9 PP, 1 M (neonate); Dam at **Rassia** N02.00021/E30.90450, 22.viii.17, 14 PP; Camp Ushudi at **Djuparam** N02.10439/E31.02308, 20.viii.17, 1 F (HLC); Camp Ushudi at **Djuparam** N02.10439/E31.02308, 22.viii.17, 3 FF (HLC). **Djuparam** N02.10439/E31.02308, Oct-Nov 2017, 3 FF (HLC).

Historical specimens from the Natural History Museum in London included: **Nigeria**, R. Assob, undated, DM Roberts, 2 FF pinned; **Sudan**, Labossi, 19.x.49, DJ Lewis, 1 F pinned; **Sudan**, Gyagya, 19.x.49, DJ Lewis, 1 F pinned; **Ethiopia**, R. Karsa near Jimma, 25.x.73, 2 FF pinned; **Ethiopia**, R. Menkenya, 12.i.84, HBN Hynes, 1 M slide; **Tanzania**, Korogwe area, 10.i.69, JN Raybould, 1 F 'around man' pinned; **Tanzania**, W. Usumbaras Shume, 08.vi.64, 1 L spirit; **Kenya**, Kitale Cherangani, 03.ii.63, RG Highton, 1 F pinned; **Kenya**, R. Cheptabrara Nyonza, Jan 1961, RB Highton, 1 L spirit; **Uganda**, Lule Bugishu, 01.i.42 EG Gibbins, 1 F pinned; **Uganda**, Sifi, 03.iv.32, EG Gibbins 1 P slide; **Uganda** Mt Nkokonjeru, 01.iv.32, EG Gibbins, 3 P slide; **Uganda**, R. Kasala, nr Mukono, 05.xi.32, EG Gibbins, 1 P slide; **South Africa**, KZN Ngoteke waterfall, 06.x.00, RW Crosskey, 1M slide; *S. dentulosum* **lectotype**, 1 F pinned; *S. dentulosum* **paralectotypes**, 2 FF pinned; *S. ruwenzoriensis* **paratype**, Uganda, Bwambe Pass, 06.i.32, EG Gibbins, 1 P slide; *S. gilvipes* **holotype**, 1 M pinned; **S. gilvipes paratype**, Cameroon, Baugase, 1 F slide; *S. bisnovem* **paratype**, 1 F pinned; *S. kauntzeum* **paratype**, 1 F pinned.

Historical specimens from the Royal Museum for Central Africa in Tervuren included: **DRC**, R. Ituri-Kibali R. Djuda, 1950, A Fain, 4 FF/PP; **DRC**, Ituri R. Tse, 1950, A Fain, 2 FF/PP, 4 MM/PP; **Rwanda**, Rugege Forest—Clairiere Kamobuga, Sep-Nov 1949, A Fain, 5 FF (HLC); **Rwanda**, Rugege Forest–R. Muruhondo, 05.ix.49, A. Fain, 2 FF/PP, 3 MM/PP.

Other historical specimens included: **Uganda**, R. Awadu at waterfall 03.6283˚ N 31.8900˚ E, 12.ix.13, RJ Post, 2PP + 2LL.

## Molecular detection of *O. volvulus* infection in potential vectors

Appetitive adult female blackflies caught on human bait were first identified to species using morphological characters (see above). No attempt was made to dissect flies to determine their parity-status, and as a result, flies which were screened for *O. volvulus* and *O. ochengi* DNA included both parous and nulliparous flies without distinction. Samples of specimens which were screened are listed in Table 1.

After identification, single adult blackflies were divided into body and head under a stereo dissecting microscope (Zeiss) using RA lamb embryo dishes (Fisher Scientific), watchmaker forceps (VWR) and dissecting needles (Carl Roth). The chance of cross contamination by parasite DNA during this process is negligible because any parasites will remain within the alcohol-fixed tissues within the undamaged cuticle of the fly. However, to prevent possible

**Table 1.** *Simulium* samples screened for *Onchocerca* DNA in head and body.

| Site | Lat° Long° | Date | Number Specimens Screened[1] | |
|---|---|---|---|---|
| | | | *S.dentulosum* | *S.vorax* |
| Djuparam | N2.10439 E31.02308 | Feb-Mar 2018 | 37 | 1 |
| Dabu | N2.07365 E31.02237 | 08-04-2018 | 2 | 1 |
| Kulubu | N2.09487 E31.00255 | 07-04-2018 | 5 | 0 |
| Djuparam | N2.10439 E31.02308 | April 2018 | 1 | 0 |
| Kulubu | N2.09487 E31.00255 | 05-04-2018 | 12 | 0 |
| Djuparam | N2.10439 E31.02308 | Oct-Nov 2017 | 15 | 1 |
| Djuparam | N2.10439 E31.02308 | Oct-Nov 2017 | 16 | 0 |
| Djuparam | N2.10439 E31.02308 | March 2018 | 26 | 1 |
| Djuparam | N2.10439 E31.02308 | March 2018 | 38 | 0 |
| R. Koda at Rassia Dam | N2.00021 E30.90450 | 22-08-2017 | 1 | 0 |
| R. Koda at Rassia Dam | N2.00021 E30.90450 | 22-08-2017 | 1 | 0 |
| R. Koda at Rassia Dam | N2.00021 E30.90450 | 22-08-2017 | 1 | 0 |
| **TOTAL** | | | **155** | **4** |

[1]Specimens are adult females caught at human bait.

carryover of fixed tissue fragments from the torn neck, the dissection tools were washed with 70% EtOH and deionized water between specimens. Body and head were individually and separately transferred into 2 ml Precellys Ceramic Kit 1.4 mm tubes (VWR) containing 80 µl PBS. The samples were disrupted with a Precellys 24 homogeniser (VWR) at 5500 rpm (3x30 s) before DNA was extracted using the QIAGEN QIAamp DNA Mini Kit Tissue protocol (Qiagen). Incubation at 56°C was extended overnight and samples were eluted in 50 µl kit elution buffer.

To detect PCR inhibiting factors, all extracted DNA was tested in a PCR reaction spiked with $10^4$ copies/µl of a plasmid containing the murine interferon-γ gene (GeneBank: AC_000032.1) as described by Colebunders et al. [61]. An inhibitor was considered to be present if the spiked plasmid signal differed more than 3 cycles from the no-template control. In such a case, the sample DNA was diluted until the inhibitory effect was removed, usually 1:10 or 1:100.

DNA from single heads and bodies was then screened for *O. volvulus* using a qualitative real-time triplex PCR that also differentiates *O. volvulus* from *O. ochengi*. Although both are closely related, they vary in the NADH dehydrogenase subunit 5 (*ND5*) gene so that species-specific probes could be used (GenBank: AY462885.1 and FM206483.1). In addition, the genus-specific mitochondrial *16S* rDNA gene was included in the triplex PCR. A total volume of 20 µl reaction mixture comprised of 2 µl DNA, 1X HotStarTaq Buffer (Qiagen), 4.5 mM $MgCl_2$, 200 µM dNTPs (50 µM each, Fisher Scientific), 0.5 Units HotStarTaq (Qiagen) and primers and probes listed in Table 2. The PCR was performed using a Rotor Gene Q (Qiagen) with the following cycling conditions: *Taq* polymerase activation at 95°C for 15 min, 45 cycles

**Table 2. Primer and hybridization probe sequences used in the triplex PCR.**

| Name‡ | Primer and probe sequences | Concentration (nM) |
|---|---|---|
| OvOo ND5 FW | 5'- GCTATTGGTAGGGGTTTGCAT-3' | 300 |
| OvOo ND5 Rev | 5'- CCACGATAATCCTGTTGACCA -3' | 300 |
| 16S FW | 5'- AATTACTCCGGAGTTAACAGG -3' | 500 |
| 16S Rev | 5'- TCTGTCTCACGACGAACTAAAC -3' | 500 |
| Ov probe | TQP 5' 6-Fam TAAGAGGTTATTGTTTATGCAGATGG 3' BHQ-1 | 50 |
| Oo probe* | TQP 5' Hex TAAGAGATTGTTGTTTATGCAGATAGG 3' BHQ-1 | 50 |
| 16S probe | TQP 5' Cy5 TACAACATCGATGTAGCGCAGC 3'BBQ-650 | 75 |

‡ Ov = *Onchocerca volvulus*, Oo = *Onchocerca ochengi*.

of 95˚C for 10 s and 61˚C for 30 s with fluorescence acquisition on the Fam, Hex and Cy5 channels. In every run, plasmids containing the respective gene sequences were included as positive controls [62]. A sample was considered positive for a gene if at least one of the three replicates produced a signal while using the 15% outlier removal setting in the Rotor Gene Q Software (version 2.3).

## Dna barcoding and molecular phylogeny of potential vectors

DNA extracted from specimens which had been identified morphologically as *S. dentulosum* and *S. vorax*, and had been sampled for detection of *Onchocerca* DNA (see above), was used for DNA barcoding of the blackflies (Table 3). For this, the mitochondrial cytochrome *c* oxidase subunit I (COI) *locus* was amplified using the primers LCO1490 (5'-GGTCAACAAAT-CATAAAGATATTGG-3') and HCO2198 (5'-TAAACTTCAGGGTGACCAAAAAATCA -3') [63] and GoTaq Hot Start Green Master Mix 2X (Promega, Madison, WI). Amplification conditions consisted of a first denaturation step at 95˚C for 5 min followed by 30 cycles of amplification (94˚C for 30 s, 50˚C for 30 s and 72˚C for 1 min) and a last step of 7 min at 72˚C. Amplicons were subsequently sequenced on both strands by Genoscreen (Lille, France) using the same primers. The sequences were edited using Geneious Pro software (version 9.0.5) [64] and deposited in GenBank under accession numbers MT323166-MT323206.

The obtained mitochondrial sequences, together with reference sequences of other *Simulium* species from GenBank, were used to construct a molecular phylogeny. Phylogenetic constructions were carried out using Bayesian Inference analyses with the software MrBayes version 3.2.3 [65]. The model of sequence evolution that best fit the data was determined jModelTest version 2.1.4 [66] based on the Akaike Information Criterion (AIC). The Bayesian analyses consisted of two independent runs of four incrementally heated Metropolis Coupled Markov Chain Monte Carlo (MCMCMC) starting from a random tree. MCMCMC was run for 2,000,000 generations with trees and associated model parameters sampled every 100 generations. The convergence level was validated by an average standard deviation of split frequencies inferior to 0.05. The initial 10% of trees from each run were discarded as burn-in and the consensus phylogeny and posterior probabilities were obtained from the remaining trees. Finally, the phylogeny was visualized and rooted to midpoint using FigTree version 1.4.2 [67].

## Results

### Field collection of simuliidae

The numbers of *S. neavei* caught by HLC from the Nyagak-Bondo focus in 2011 have already been published [37], but in addition four *S. vorax* were also collected in November 2011. All

**Table 3. Collection data for specimens used for DNA barcoding[1].**

| Specimen | [2]Clade / Subclades | Table 3 Code | Year | [3]PCR *Oncho spp.* |
|---|---|---|---|---|
| 1 | A | D66 | 2017 | OvD |
| 2 | A | D84 | 2017 | Negative |
| 3 | A | D88 | 2017 | Negative |
| 4 | A | D90 | 2017 | Negative |
| 5 | A | D92 | 2017 | Negative |
| 6 | A | D95 | 2017 | Negative |
| 7 | A | D129 | 2018 | Negative |
| 8 | A | D134 | 2018 | Negative |
| 9 | A | D137 | 2018 | Negative |
| 10 | A | D147 | 2018 | Negative |
| 11 | A | D149 | 2018 | Negative |
| 12 | B | D68 | 2017 | Negative |
| 13 | B | D72 | 2017 | Negative |
| 14 | B | D83 | 2017 | Negative |
| 15 | B | D91 | 2017 | Negative |
| 16 | B | D125 | 2018 | Negative |
| 17 | B | D126 | 2018 | Negative |
| 18 | B | D135 | 2018 | Negative |
| 19 | B | D139 | 2018 | OvV |
| 20 | B | D146 | 2018 | Negative |
| 21 | SdC | D70 | 2017 | OvD |
| 22 | SdC | D74 | 2017 | OoD |
| 23 | SdC | D85 | 2017 | Negative |
| 24 | SdC | D86 | 2017 | Negative |
| 25 | SdC | D93 | 2017 | Negative |
| 26 | C | D94 | 2017 | Negative |
| 27 | C | D127 | 2018 | Negative |
| 28 | SdC | D124 | 2018 | OoD |
| 29 | SdC | D128 | 2018 | Negative |
| 30 | SdC | D130 | 2018 | Negative |
| 31 | SdC | D133 | 2018 | OvV |
| 32 | SdC | D136 | 2018 | Negative |
| 33 | SdC | D142 | 2018 | OvV |
| 34 | SdC | D145 | 2018 | Negative |
| 35 | SdC | D148 | 2018 | Negative |
| 36 | SdC | D150 | 2018 | Negative |
| 37 | SdC | D151 | 2018 | Negative |
| 37 | SdC | D153 | 2018 | OoD |
| 38 | SdC | D156 | 2018 | Negative |
| 39 | SdC | D158 | 2018 | Negative |
| 40 | SdV | D64 | 2017 | OvD |

Notes

[1]All barcoded specimens were collected by the HLC method from Djuparam (N2.10439˚, E31.02308˚), within 5km from the River Kakoi.

[2]Sd = *S. dentulosum* clade and subclades (A, B, C); Sv = *S. vorax* clade (see Fig 3)

[3]PCR positive results: OoD = *O. ochengi* infected specimens; OvD = *O. volvulus* infected specimens; OvV = *O. volvulus* infective specimens.

other vector collections (river prospections and human landing catches) from all years are mapped in Fig 2A and 2B. Lists of all blackfly species identified from rivers (mostly from pupae, with occasional examination of larvae) and Human Landing Catches (HLCs) from the Kakoi-Koda focus and nearby sites in 2015, 2017 and 2018 are presented in S1 and S2 Tables. The results from the crab trapping sites within the Kakoi-Koda focus are presented in S3 Table. A list of collections in 2009 and 2016 aimed only at vector species (river prospections including crab trapping, and spot checks for HLCs) from DRC and Uganda around the outside of the Kakoi-Koda focus are presented in S4 Table.

No specimens of *S. damnosum* s.l., *S. albivirgulatum*, or *S. woodi* were collected from any rivers or amongst any HLCs from either DRC or Uganda.

Crabs collected from within the Kakoi-Koda focus in 2015 and 2016 were not found to be infested with *S. neavei* (zero infested crabs amongst 97 examined). However, a single collection made in October 2009 from the River Kakoi (in the middle of the focus) was heavily infested (11 crabs infested out of 12 examined), although three other sites within the Kakoi river system were negative, and so were two sites in the adjacent Ori river system (just outside the focus), and no adult flies were caught by any of the accompanying spot-checks. HLCs from within the focus in 2017 and 2018 included no *S. neavei*, but significant numbers of *S. dentulosum* and *S. vorax* were collected.

Crabs collected from outside the Kakoi-Koda focus in Uganda in October 2009 (River Nyagak and tributaries) were found to be heavily infested (67 crabs infested out of 283 examined, with only 2 negative sites out of seven surveyed). However, rates of infestation amongst crabs collected from DRC also outside the Kakoi-Koda focus in October 2009 and May/June 2016 were generally zero, except for a cluster of significant infestation in three adjacent rivers (the Rivers Omi, Ake and Mi) which are a little more than 30 km NW of the Kakoi-Koda focus, and a few *S. neavei* were caught at HLCs associated with these rivers (12 adult females from one site).

## Molecular detection of *O. volvulus* in anthropophilic blackflies

The results of the screening of specimens for *O. volvulus* and *O. ochengi* DNA in heads and bodies of individual anthropophilic blackflies are presented in Table 4.

Only four specimens of *S. vorax* were available for screening and one of them (25%) was positive for *O. volvulus* in the body (but not the head), and none of them were positive for *O. ochengi*. The percentage of infection in *S. dentulosum* was 30% infected and 11% infective for *O. volvulus*, and 5% infected for *O. ochengi*. Therefore, it is highly probable that *S. dentulosum* is a vector of human onchocerciasis in the Kakoi-Koda focus, and probably a vector of bovine onchocerciasis as well. The situation is less certain for *S. vorax*.

## Morphotaxonomy of *S. dentulosum*

A formal taxonomic description of *S. dentulosum* found breeding and biting humans in the Kakoi-Koda focus is available in S1 Text.

**Table 4. Infection and infectivity rates for *O. volvulus* and *O. ochengi* amongst anthropophilic blackflies within the Kakoi-Koda focus.**

| Species | Sample size | Ov-infected | Ov-infective | Oo-infected | Oo-infective |
|---|---|---|---|---|---|
| *S. vorax* | 4 | 1 (25%) | 0 (0%) | 0 (0%) | 0 (0%) |
| *S. dentulosum* | 155 | 47 (30%) | 17 (11%) | 8 (5%) | 0 (0%) |

**Notes:** Infective specimens are those with parasite DNA in the head, and infected flies are those which were positive for parasite DNA in the heads and/or the bodies.
Four specimens of *S. dentulosum* were positive for *O. volvulus* in both head and body, and four specimens were co-infected with *O. volvulus* (one of them infective) and *O. ochengi*.

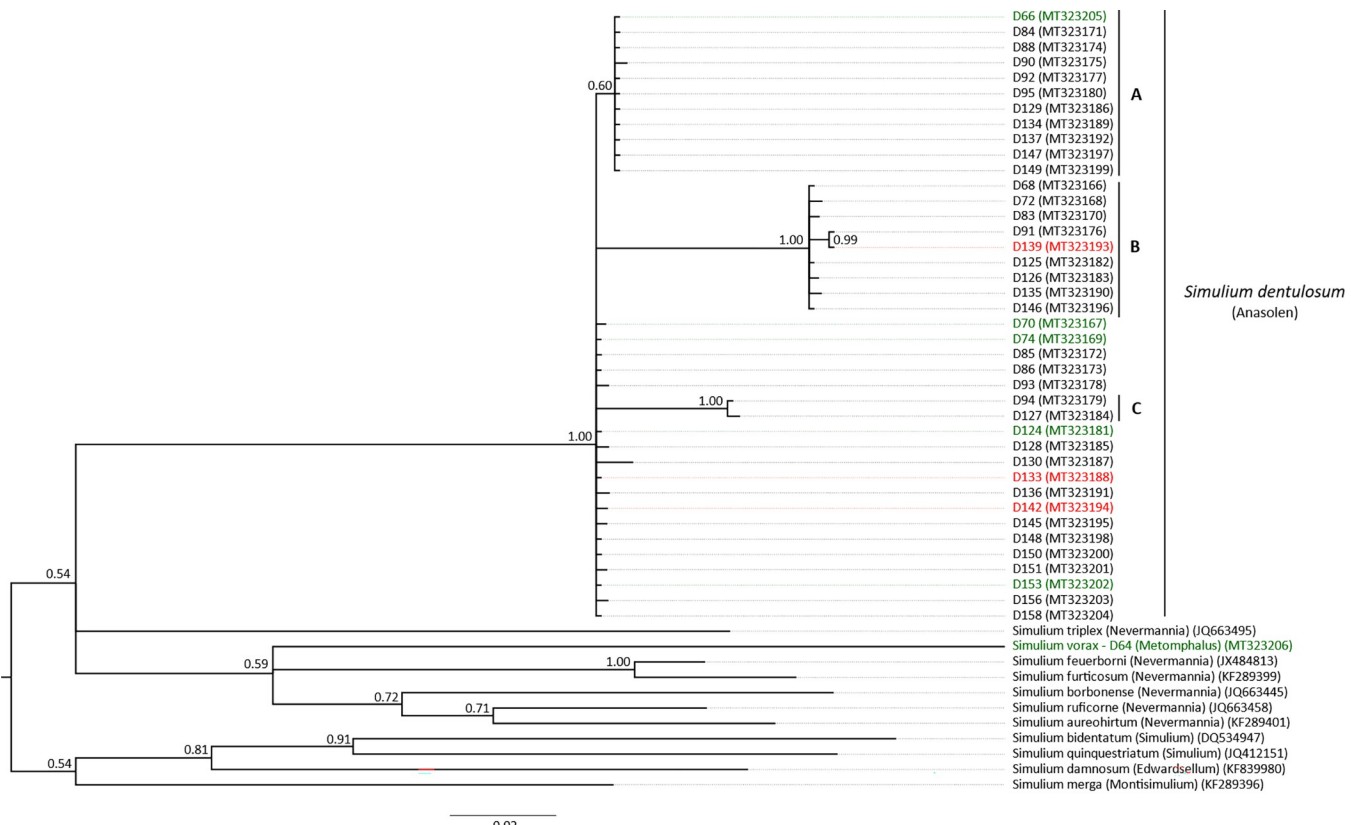

**Fig 3. Phylogenetic tree based on the mitochondrial sequences (COI, 658 bp) of specimens identified as *Simulium dentulosum* and *Simulium vorax* from Kakoi-Koda Focus.** The analysis was performed with Bayesian inference under the GTR + I + G model of sequence evolution. The green and red colours indicate specimens identified as "infected" by either *O. volvulus* and *O. ochengi* and "infective" for *O. volvulus*, respectively. The sequenced *S. vorax* specimen was infected with *O. volvulus*. Posterior probabilities are indicated at the nodes. GenBank accession numbers of sequences are indicated between brackets.

The morphotaxonomic comparisons indicated that *S. dentulosum* is quite variable between populations across Africa. For example, larval specimens from the Western Usambara mountains in Tanzania were found to have a very distinctive hypostomium (with the usual nine teeth, but the apical margin was triangular and projected forward, with the point of the triangle under the median tooth), the cephalic apotome markings of the larval head were very distinct in some populations, but quite indistinct in specimens from Kenya. Last instar larvae varied markedly between localities in size and in general dark/light colouration of the cuticle. The surface sculpturing of the pupal gills and the thoracic dorsum was also found to show marked variation. For example, the dorsum of a male pupa from Ngoteke in South Africa had a dense covering of pointed tubercles, whereas specimens from various sites in Uganda (Sii, Mt Nkokonjeru and R. Kasala) all had less dense round tubercles. *Simulium ruwenzoriensis* (paratype) had a dense covering of mostly round tubercles, with some pointed. The ventral plate of males from the River Menkenya in Ethiopia had large pointed shoulders, and larvae from some localities were much darker than others. The density of hairs and scales over the head, thorax and abdomen of female adults varied markedly between localities, as did the size of the basal tooth of the claw (as already indicated by Freeman & de Meillon [43]).

The number of pupal gill filaments in specimens collected from the Kakoi-Koda focus was usually (but not always) 15, instead of the normal 14 (made up of seven pairs). In all gills examined with 15 filaments the additional one was the result of the posterior inner 'pair' being a

triplet. Of 24 specimens examined for both pupal gills, 21 specimens had 15 filaments on both sides, one specimen had 14 on both sides, two had 15 on one side and 14 on the other, and one specimen had 15 on one side and 16 on the other (an additional inner 'pair' was manifest as a triplet). In addition, a few specimens had stubs (very short filaments), for example one damaged specimen (with only one gill) had the basic 14 filaments with a 15th filament in the normal place but reduced to a stub. A gill from another specimen had 15 filaments, but the triplet had the stub of a sixteenth filament.

The only feature that was unique and consistent (or almost consistent) to the contemporary anthropophilic populations of *S. dentulosum* from the Kakoi-Koda focus was the bare pleural membrane, although a single female specimen examined did have a single golden hair on one pleural membrane, and two out of three neonate males examined had distinctly hairy pleural membranes. All other (historical) female specimens examined had some golden hairs on the pleural membrane (including other *Anasolen* species, except *S. kauntzeum* but including *S. bisnovem*, which had 4 or 5 hairs). The specimens of both sexes of *S. dentulosum* collected by Fain in 1950 were also found to have distinctly hairy pleural membranes, but he did not record them as anthropophilic, and they were collected 40km west of the Kakoi-Koda focus [34].

### DNA Barcoding of *S. dentulosum* and *S. vorax*

The amplification of the mitochondrial COI *locus* from the identified blackfly specimens produced 41 sequences of 658 nucleotides without deletions or insertions (*S. dentulosum*, n = 40; *S. vorax*, n = 1). The mitochondrial phylogeny (Fig 3) shows that all of the sequences from *S. dentulosum* group together in one major well-supported clade (Posterior probability (PP) = 1.0), and as expected, the single sequence obtained from *S. vorax* did not fall within the *S. dentulosum* clade.

Within the *S. dentulosum* clade, there are three obvious subclades, A, B and C, supported by PP = 0.60, PP = 1.00 and PP = 1.00, respectively (Fig 3). This topology indicates that the identified *S. dentulosum* are represented by different mitotypes occurring in sympatry and highlights significant intra-species variation. It is noteworthy that within the *S. dentulosum* clade, specimens infected with *O. volvulus* do not fall into any specific group since they are present at the base of the main clade (D70, D74, D124, D142 and D153) and also in the subclade B (D139). Finally, uninfected specimens are distributed along the major clade and subclades.

## Discussion

### Taxonomic status of *S. dentulosum* and *S. vorax* in the Kakoi-Koda focus

The subgenus *Anasolen* is restricted to the Afrotropical region (including the Arabian Peninsula) and consists of 11 named species, including *S. dentulosum*. These are typical of mountainous regions and are mostly restricted to particular mountain blocks in East and East-Central Africa, although *S. dentulosum* is common and widespread (in 22 countries) and extends its ecological range from highland areas into clear, swift lowland streams [60, 68]. *Simulium dentulosum* does not regularly bite humans, except in a very few cases [5, 43, 52]. Fain [34] recorded *S. dentulosum* breeding in the Ituri highland (40 km from the Kakoi-Koda focus; Fig 2A) but he did not record it biting humans, although he did record *S. dentulosum* as being attracted to humans some 200 km southwest along the rift valley near Mutwanga and also biting humans in the Rugege Forest in Rwanda [55]. However, there is no mention of *S. dentulosum* being an onchocerciasis vector in that region. In short, there seems to have been no previous suggestion that *S. dentulosum* (or any other species belonging to the subgenus *Anasolen*) might be a vector of human onchocerciasis.

Morphologically, *S. dentulosum* is a rather variable species across Africa, and the contemporary anthropophilic population from the Kakoi-Koda focus is not exceptional within this range of variability. It is especially significant that the male and female genitalia are more or less typical of *S. dentulosum*. The only morphological features which stand out from the species description are the bare pleural membrane of contemporary anthropophilic females within Kakoi-Koda focus and the presence of 15 instead of 14 pupal gill filaments. However, the bare pleural membrane is not a powerful character as Crosskey [54] considered that the presence of hairs varied at the subgenus level, and sometimes even within *Anasolen* species. Freeman and de Meillon [43] described such variations between populations of *S.* (*Anasolen*) *debegene*, and the presence/absence of hairs on the pleural membrane of specimens of *S.* (*Anasolen*) *bisnovem*, and *S.* (*Anasolen*) *kauntzeum* examined in this study was contrary to the respective species descriptions by Freeman & de Meillon [43]. In any case a single female specimen of *S. dentulosum* from the Kakoi-Koda focus had a single hair on one pleural membrane. It is also possible that some (or all) of the hairs were rubbed off from some specimens caught through human landing catch. Regarding the gill filaments, it is clear that these populations (and others recorded nearby by Fain [34]) are exceptional amongst *S. dentulosum* in that they have 15 gill filaments. However, this is not the case for all specimens (both historical and recent) caught within or near the current-day boundaries of the Kakoi-Koda focus. Specimens were found with 14 or 15 gill filaments on both gills, and various sorts of intermediates have been found as well as one gill with 16 filaments. It is clear that the number of gill filaments cannot be interpreted as reliable evidence that these populations are comprised of a species distinct from *S. dentulosum*. In summary, there is no strong morphotaxonomic evidence to consider the anthropophilic population of *S. dentulosum* in the Kakoi-Koda focus to be a new species or subspecies. This conclusion, of course, still leaves open the possibility that they could represent an endemic cryptic species which might be revealed by further molecular studies or by cytotaxonomy. This general possibility is well documented and quite common in blackflies [69], but at present there is no such evidence for this in *S. dentulosum*.

*Simulium vorax* has been previously recorded biting humans (usually in the presence of cattle, which are its preferred blood-host), and it has been shown to support development of *O. volvulus* in the laboratory [70]. Furthermore, it belongs to the subgenus *Metomphalus* and hence it is closely related to *S. albivirgulatum* (a known vector in central DRC). Therefore, it is not a great surprise to find that this species is anthropophilic in the Kakoi-Koda focus, and there is no reason to question the taxonomic status of the Kakoi-Koda specimens because they were all easily identified using the standard characters described by Freeman and de Meillon [43] and Crosskey [54].

## Potential vectors in the Kakoi-Koda focus

The presence of *O. volvulus* DNA in heads as well as bodies of *S. dentulosum* is strong evidence that *S. dentulosum* is a vector (possibly the only current-day vector) in the Kakoi-Koda focus. It is difficult to see how *O. volvulus* DNA could be detected in the head of the blackfly if the parasite larvae were not developing to the L3 stage (although this does not guarantee that the L3 larvae are viable). It is also unlikely that a significant number of the wild-caught blackflies which were positive for *Onchocerca* DNA could be the result of cross-contamination. Not only does the method by which specimens were handled, and the DNA extracted, make this unlikely, but it is difficult to see why bodies would show higher rates of contamination than heads, and positive specimens did not occur in runs (i.e. the next specimen processed after a positive specimen did not have a raised chance of also testing positive). It is possible that torn shreds of *O. volvulus* microfilariae might sometimes persist on the cibarial armature within a

blackfly head after feeding upon an infected person, but this cannot be the case for *S. dentulosum* because it has an unarmed cibarium [54], and so it is difficult to see how *O. volvulus* DNA comes to be in the head except through the migration of infective L3 parasites into the head, thus supporting the conclusion that *S. dentulosum* is a vector.

Insufficient numbers of specimens were screened to be certain about whether *S. vorax* may or may not be a vector in the Kakoi-Koda focus, but it is possible, because *O. volvulus* DNA was detected in the body (not head) of one of the four specimens screened. The presence of *O. volvulus* DNA in the abdomen does not prove that the infection is viable (or even that the parasites have successfully escaped the gut). However, Wegesa [70] has shown that *S. vorax* can support the development of *O. volvulus* in the laboratory in Tanzania, and therefore it remains possible that *S. vorax* is also a vector, but this cannot be ascertained without further studies.

It is unlikely that *S. albivirgulatum* is present and transmitting onchocerciasis in the Kakoi-Koda focus, not only because it was not found during our study, but also because it has never been recorded anywhere near Ituri Province. Furthermore, the only focus where it is a known vector is in the 'cuvette centrale' of DRC, which is a low-altitude plain in the middle of the equatorial rain forest [71] where *S. albivirgulatum* breeds in large slow-flowing rivers, quite unlike the Ituri Highlands.

It is also unlikely that *S. damnosum* s.l. was biting and transmitting onchocerciasis, but was undetected, because of the significant effort that was put into searching for the breeding sites in all seasons of the year, as well as its absence from HLCs. *Simulium damnosum* is a complex of cryptic species (sometimes referred to as sibling species) which show a very wide range of ecological preferences, but in east and central Africa most of them do not bite humans [39]. Fain [34] recorded *S. damnosum* s.l. from the Ituri Highlands a little more than 30 km SW of the Kakoi-Koda focus, and he specifically stated that they did not bite humans. Further to this, the Nkusi form of *S. damnosum* s.l. has been identified chromosomally in Uganda from the Nyagak River [5] which has its source in DRC close to the Kakoi-Koda focus, but the Nkusi form is a non-anthropophilic cytospecies in Uganda [39].

*Simulium neavei* was not captured during HLCs within the focus, and it was only found infesting crabs in one collection in one year (2009) at one site within the focus, although many crabs were examined at many sites over several years. However, there are strings of onchocerciasis foci perched along the highlands on either side of the Rift Valley [72], and all of them are transmitted by *S. neavei* [5, 29, 36]. Along the eastern side, above Lake Albert and more or less opposite the Kakoi-Koda focus lie the Itwara, Mpamba-Nkusi, Wambabya-Rwamarongo and Budongo foci in Uganda, and the Kakoi-Koda focus is apparently the most southerly of a similar string of foci, along the highlands to the west of the Rift Valley, which continue northward across the border in Uganda with the Nyagak-Bondo, Maracha-Terago and West Nile foci [73] (see Fig 1). Furthermore, we found *S. neavei* biting people NW of the Kakoi-Koda focus near the Rivers Omi, Ake and Mi, where we also been found it breeding. *Simulium neavei* seems to be a very efficient vector and highly anthropophilic [74], and hence it might be able to maintain a focus even at low annual biting rates, and Prentice [36] reported that "*adult flies are rare at most seasons*" within the *neavei*-transmitted Nyagak-Bondo focus in Uganda. Onchocerciasis vectors of the *S. neavei* group are generally considered to be strongly associated with heavily-shaded streams and rivers passing through forest or woodland, to the extent that deforestation can lead to the disappearance or at least severe reduction in population density [52, 75, 76]. This effect is so marked that the very first successful elimination of any onchocerciasis vector from any focus was *S. neavei*, eliminated by 'bush clearence' in Riana focus in Kenya in 1943 [77]. Whilst *S. neavei* breeding sites are found under dense vegetational cover, Mpagi et al. [73] described how biting was most common near the forest edge and occurred up to 4 km away from the forest at Itwara focus in Uganda. In summary, the evidence is strong that *S.*

*dentulosum* is acting as the main vector in the Kakoi-Koda focus, but in view of the known habits of *S. neavei*, it would be unwise to rule out the possibility that it could also be significant in the current-day or historical epidemiology of the focus.

We were unable to confirm the existence of *S. woodi* in Ituri Province, because it was not identified from any of our samples. Pupae and larvae are very difficult (if not impossible) to separate from *S. neavei*, but adult females are very easy to identify, and were not found amongst HLCs. There is no doubt that the small number of adult females which were collected outside of the Kakoi-Koda focus and identified as *S. neavei* were correctly identified, and if *S. woodi* does occur in the study area, it is unlikely that it is a significant vector.

## The origin and history of the Kakoi-Koda focus

It is probably too late to try to reconstruct the history of onchocerciasis in the Kakoi-Koda focus with any degree of certainty, but the identity of the probable current vector (*S. dentulosum*) is most unusual and warrants some discussion of how this might have come about. Firstly, it is important to point out that there is no historical evidence concerning the age of the focus. Fain and Hallot [3] claimed that Haut-Ituri was free of onchocerciasis, but Fain did not collect Simuliidae from the NW of the Ituri Highlands, and it is unclear how they came to their conclusion, and in any case it is entirely possible that they would not have considered the scarp face of the Ituri Highlands to be part of 'Haut-Ituri'. The first evidence that we have for the existence of the Kakoi-Koda focus comes from the REMO surveys carried out from 1997 to 2008, and from these surveys we know the focus is at least 25 years old, but we do not know how much older it might be. It is entirely possible that this is a relatively new endemism in an area of increasing population density, and resulting from the bringing together of an unusual anthropophilic population of *S. dentulosum* (anthropophilic as a result of an unusual genetic composition, or because of the unique and changing environment) with immigrant infected people. However, the geographical position of the Kakoi-Koda focus is similar to that of the Nyagak-Bondo focus (and other foci further north along the edge of the Rift Valley) just over the border in Uganda where *S. neavei* is the vector, and therefore it is entirely plausible that the focus is simply the most southerly of a string of foci along the western ridge of the Rift Valley, all with a similar long-standing well-wooded ecology. Under this scenario, the Kakoi-Koda focus could have been established long ago (for example, as a result of the movement of the Luo people from South Sudan in the 16th century), and it is likely that the historical vector could have been *S. neavei*, which only declined fairly recently due to the anthropogenic removal of tree cover and changes of land-use, as has been well documented in other foci [28]. In this scenario, the collection of infested crabs from the Kakoi-Koda focus in 2009 would have represented the last remnants of a declining population. The same ecological changes could also have resulted in increased contact between humans and *S. dentulosum* and consequently the use of humans as blood hosts (with or without genetic adaptation). The Ituri Highlands is a significant biodiversity hot-spot with an above average proportion of endemic species [78]. It is therefore possible (but not essential to this scenario) that the populations of *S. dentulosum* endemic to this particular mountain block were genetically unusual with a higher potential for anthropophily than other populations of the same species elsewhere. It remains possible that *S. dentulosum* was always the vector in the Kakoi-Koda focus, but even if it has replaced a previous historical vector, that does not mean that it will be immune to the effects of further deforestation. *Simulium dentulosum* is strongly orophilic (mountain-loving) and mountains are often associated with high rainfall and closed-canopy forests with shade. However, the breeding site preferences of *S. dentulosum* are generally not well understood and it has been recorded in drier less shady places and it does not usually extend into lowland forest.

Continuing anthropogenic change (including removal of tree cover) in the Kakoi-Koda focus will also affect hydrochemistry, microclimate and seasonality of river discharge, and it is unclear whether this might render the area unsuitable for *S. dentulosum* in the future.

## Is the Kakoi-Koda focus isolated?

The possibility that this could be a focus of infection without local transmission, but being sustained by the immigration and settlement of people who have become infected elsewhere, can be excluded because infections have been found in children who have no history of living anywhere else [11]. However, the epidemiology of infection and progress towards elimination during interventions, such as CDTI, could still be influenced by immigration of infected people or infected vectors from other foci (and vice versa).

The movement of people is outside the scope of this report, but there are two sorts of evidence we can consider. Firstly, the geographical distribution of infection in humans. If vectors and/or humans are moving around in significant numbers and all other things being equal, we would expect to see a more or less continuous distribution of infection, but this is not what we see. The REMO results [8, 9] from the Ituri Highlands show a concentration of hyper-endemism in one corner (the Kakoi-Koda focus), with hypo-endemism in populated areas between the Kakoi-Koda focus and other adjacent hyper-endemic areas. The sources of the River Kakoi and its tributaries flowing through the Kakoi-Koda focus are mostly very close to those which flow through the Ugandan Nyagak-Bondo focus (in some cases just 1 km apart) and the DRC endemic area to the NW (centred on the Omi, Mi and Ake rivers). These source areas are sometimes marshy and unsuitable for vector breeding, but the distances between the endemic areas are well within the dispersal range of *S. damnosum* s.l. vectors, and hence the low levels of endemicity in these areas suggest that *S. dentulosum* (assuming it is the vector), like *S. neavei*, does not have a strong tendency to disperse. This is supported by the absence of emigrant anthropophilic *S. dentulosum* being caught in the adjacent endemic areas which are known around the Kakoi-Koda focus, and the absence of immigrant *S. neavei* in HLCs within the Kakoi-Koda focus. The occurrence of *S. vorax* in HLCs near the River Nyagak in Uganda could be the result of local breeding rather than immigration from DRC, because McCrae [79] found *S. vorax* to be one of the most common blackflies breeding in that part of Uganda. Dispersal patterns of *S. dentulosum* (and *S. vorax*) are completely unknown, but in the Itwara focus in Uganda *S. neavei* has a dispersal range of only about 4 km [73] and this lack of dispersal meant that it was possible to apply larvicides to eliminate vector breeding from adjacent river systems one by one within the focus, without any problem of reinvasion [28]. It is this principle of low rates of vector dispersal that has allowed Uganda to treat the individual foci which constitute the two strings of *neavei*-transmitted foci along the highlands which form eastern and western ridges of the Rift Valley as separate transmission zones (that is to say the Itwara, Mpamba-Nkusi, Wambabya-Rwamarongo and Budongo foci along the eastern side, and the Nyagak-Bondo, Maracha-Terago and West Nile foci further north along the western side). Transmission has probably been eliminated from all of them [29], and this is in spite of the fact that they were often occupying adjacent river basins. It seems that the Kakoi-Koda focus could be thought of as simply the most southerly of the string of foci along the highland ridge forming the western side of the Rift Valley, and in spite of its unusual vector it is most likely to be quite isolated from neighbouring foci, and could be considered as a distinct transmission zone in its own right. Certainly, it seems that it has been possible to eliminate transmission from the adjacent Nyagak-Bondo focus [29] even while transmission continued in the Kakoi-Koda focus.

In summary, the Kakoi-Koda focus seems to be rather well isolated from vector immigration. Immigration is unlikely from the south and east, because the nearest foci are on the other side of Lake Albert (at least 40 km distant), and they are *S. neavei* foci, and *S. neavei* has a very short dispersal range. To the north, the Nyagak-Bondo focus is very close, just across the border in Uganda and *S. neavei* used to be abundant there before it was eliminated in 2014 by larviciding. In any case, the Ugandan experience is that *S. neavei* has very limited dispersal, such that it has not interfered with vector elimination in neighbouring foci. There is also an area of onchocerciasis transmitted by *S. neavei* within DRC to the NW and close to the Kakoi-Koda focus, but this is unlikely to be a source of immigrant vectors, not only because of the intrinsic low dispersal shown by *S. neavei*, but also because none were collected by HLCs within the Kakoi-Koda focus (which indicates a lack of mass immigration of *S. neavei*, as well as a lack of local breeding). Finally, to the west and southwest of the Kakoi-Koda focus lies the rest of the Ituri Highlands. In this area onchocerciasis was either absent, sporadic or at most hypo-endemic according to REMO, and vector blackfly species have not been recorded during historical or recent surveys. Therefore, it is doubtful whether there is active transmission, and doubtful whether there is any potential for the area to be a source of migratory vectors, but the current evidence is uncertain and needs to be confirmed.

## Conclusions

In conclusion, there is no compelling morpho-taxonomic evidence to consider that the contemporary anthropophilic populations within the Kakoi-Koda focus consist of any blackfly species other than *S. dentulosum* and *S. vorax*. However, the frequent existence of cryptic species within morphospecies is a common and well-documented phenomenon within blackfly taxonomy, and it remains possible that cytotaxonomic and molecular investigations could still provide evidence that these populations represent new cryptic species, although this is not supported by current evidence.

The evidence that *S. dentulosum* is a vector is based upon the discovery of *O. volvulus* DNA within heads and bodies of female specimens of *S. dentulosum* caught by HLCs, and it should be noted that the detection of parasite DNA in the heads of known vectors is considered by WHO to be sufficient evidence for transmission of *O. volvulus* during monitoring of onchocerciasis elimination programmes [26]. However, it remains uncertain whether the Annual Transmission Potential (a function of annual biting rates and infectivity rates) of this newly identified vector is sufficient to maintain the focus, and the roles (if any) played by *S. vorax* and *S. neavei* remain to be elucidated. Whilst it seems certain that *S. dentulosum* was transmitting *O. volvulus* during this study, in other foci along the Albertine Rift it is *S. neavei*-transmitted, and it is quite possible that *S. neavei* was the historical vector in the Kakoi-Koda focus, having been replaced by recent anthropogenic deforestation.

The history of the Kakoi-Koda focus, and how *S. dentulosum* came to be a vector is unclear, but in spite of this it seems that the focus is fairly well isolated from vector immigration (which might be carrying *O. volvulus*). Therefore, on the basis of current information, there seems to be no reason to apply any onchocerciasis elimination interventions other than normal CDTI, although the unique vector means that it would be wise to monitor progress towards elimination more closely than in other foci which have the classical vectors.

## Supporting information

**S1 Fig. Arial photographs of Kakoi-Koda focus (20th April 2022).**
(PDF)

**S2 Fig. 3D Relief Model of Kakoi-Koda Focus.** (available online: https://ecohealthalliance.github.io/KakoiKodaFocus/KakoiKoda_3D_Laudisoit.html).
(DOCX)

**S1 Table. Alphabetical List of Blackfly Species identified from the Ituri Highlands, September-October 2015 and August 2017.**
(PDF)

**S2 Table. List of Species Identified from the Ituri Highlands by Date and Locality.**
(PDF)

**S3 Table. Crab Trapping in the Kakoi-Koda Focus.**
(PDF)

**S4 Table. List of Collections of Vectors from outside the Kakoi-Koda focus in 2009 and 2016.**
(PDF)

**S1 Text. Taxonomic description of recent S. dentulosum from the Kakoi-Koda onchocerciasis focus.**
(PDF)

## Acknowledgments

We thank the US National Science Foundation (NSF) through Ecohealth net (https://www.ecohealthalliance.org/program/ecohealthnet) for their scientific support, the Ituri province authorities for their administrative support and the village health care workers (RECO) and guides for their assistance in carrying out the field surveys. The Uganda-DRC crossborder studies were supported by APOC, The Carter Center and SightSavers. We are also grateful to the Royal Museum for Central Africa in Tervuren and the Natural History Museum in London for allowing us access to their important collections of African Simuliidae, and for supplying us with the on-site facilities to examine the specimens. We also thank them and Frans Breteler (Herbarium Vadense) for sharing their taxonomic expertise and advice.

## Author Contributions

**Conceptualization:** Rory J. Post, Anne Laudisoit, Robert Colebunders.

**Data curation:** Rory J. Post.

**Formal analysis:** Rory J. Post, Anne Laudisoit.

**Funding acquisition:** Anne Laudisoit, Robert Colebunders.

**Investigation:** Rory J. Post, Anne Laudisoit, Michel Mandro, Thomson Lakwo, Christine Laemmer, Kenneth Pfarr, Pablo Tortosa, Yann Gomard, Tony Ukety, Claude Mande, Lorne Farovitch, David W. Oguttu.

**Methodology:** Rory J. Post, Anne Laudisoit, Pablo Tortosa, Yann Gomard, Tony Ukety.

**Resources:** Robert Colebunders.

**Software:** Rory J. Post, Anne Laudisoit, Pablo Tortosa, Yann Gomard.

**Supervision:** Rory J. Post, Anne Laudisoit, Achim Hoerauf, Tony Ukety, Uche Amazigo, Didier Bakajika, Naomi Awaca, Robert Colebunders.

**Validation:** Rory J. Post, Anne Laudisoit, Robert Colebunders.

**Visualization:** Rory J. Post, Anne Laudisoit, Pablo Tortosa, Yann Gomard, Claude Mande.

**Writing – original draft:** Rory J. Post, Anne Laudisoit.

**Writing – review & editing:** Rory J. Post, Anne Laudisoit, Michel Mandro, Thomson Lakwo, Christine Laemmer, Kenneth Pfarr, Achim Hoerauf, Yann Gomard, Tony Ukety, Claude Mande, Lorne Farovitch, Uche Amazigo, Didier Bakajika, David W. Oguttu, Naomi Awaca, Robert Colebunders.

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
