## [Decision Letter · Decision Letter 0]

16 Sep 2022

Dear Dr Colebunders,

Thank you very much for submitting your manuscript "Identification of the onchocerciasis vector in the Kakoi-Koda focus of the Democratic Republic of Congo" for consideration at PLOS Neglected Tropical Diseases. As with all papers reviewed by the journal, your manuscript was reviewed by members of the editorial board and, in this case, by two independent reviewers. The reviewers appreciated the attention to an important topic. Based on the reviews, we are likely to accept this manuscript for publication, providing that you modify the manuscript according to the review recommendations. 

Sincerely,

Prof. María-Gloria Basáñez, PhD, MSc

Academic Editor

Alvaro Acosta-Serrano

Section Editor

Reviewer's Responses to Questions

**Key Review Criteria Required for Acceptance?**

**Methods**

-Are the objectives of the study clearly articulated with a clear testable hypothesis stated?

-Is the study design appropriate to address the stated objectives?

-Is the population clearly described and appropriate for the hypothesis being tested?

-Is the sample size sufficient to ensure adequate power to address the hypothesis being tested?

-Were correct statistical analysis used to support conclusions?

-Are there concerns about ethical or regulatory requirements being met?

Reviewer #1: The objectives are clearly described and the relevant methodology is described, but please note:

Lines 430-442: A statement is needed as to the trapping frequency and specifically when crabs were collected within the years 2009, 2016, and 2017 and 2018.

Reviewer #2: The study is very well done and represents a very detailed exploration of the vectors collected in the focus. However, the fly numbers were quite low.

**Results**

-Does the analysis presented match the analysis plan?

-Are the results clearly and completely presented?

-Are the figures (Tables, Images) of sufficient quality for clarity?

Reviewer #1: The analysis and figures are clearly presented, but please note the following:

Lines 462-463: Why is the phrase “seemed to have” used in the following statement: “pupae seemed to have 15 gill filaments (instead of 14, which is expected in S. dentulosum), and adult females seemed to have a pleural membrane without hairs”. Either they had these characters or they did not. Please clarify this.

It would have been useful to confirm that the cibarium of females of S. dentulosum from the Kakoi-Koda focus actually were unarmed rather than relying on a statement drawn from the literature about the cibarium for the species in general. Either the specimens from the Kakoi-Koda focus should be examined or a caveat should be added.

Reviewer #2: Results are clearly presented and support the conclusions.

**Conclusions**

-Are the conclusions supported by the data presented?

-Are the limitations of analysis clearly described?

-Do the authors discuss how these data can be helpful to advance our understanding of the topic under study?

-Is public health relevance addressed?

Reviewer #1: The conclusions are supported by the data and relevant public health issues are are addressed. Data limitations are presented. However, please note:

Lines 76-78 (also lines 98-100): “this raises the possibility that other blackfly species which are not generally considered to be anthropophilic vectors might become vectors under suitable conditions.” This statement is problematic. The authors’ Discussion and Conclusion sections about this possibility are fine, but the summary statements here are ambiguous. Are the authors suggesting that vector status of S. dentulosum and other species not currently known as vectors have recently acquired (or could acquire) the capability to become vectors or that the capability has long been present and could manifest only if environmental conditions change. Or perhaps they are suggesting something else? Regardless, clarification is needed. 

An iron-tight case that S. dentulosum is a vector of O. volvulus is not possible based on the data, despite WHO's claim that detection of parasite DNA in the heads of known vectors [S. dentulosum is not a known vector] is considered sufficient evidence for transmission. Authors, therefore, might soften the statements that S. dentulosum is a vector and recognize it as a probable vector.

Reviewer #2: yes.

**Editorial and Data Presentation Modifications?**

Reviewer #1: Given the significance of the findings—the novel discovery that S. dentulosum is a vector (or probable vector) of Onchocerca volvulus—the title emphasizing “identification” seems pedestrian.

Line 918: “the use man” should be “the use of humans”?

The term “cryptic species” has largely replaced “sibling species”, the terms having slightly different meanings.

Please note that diseases are not transmitted, but rather the agents that cause diseases are transmitted. Thus, phrases such as “onchocerciasis vector” and “vector of onchocerciasis” are incorrect.

The use of the phrases “was found to be”, “was found to show”, “was found to have”, and similar variants throughout the manuscript makes for tedious reading and should be deleted.

Reviewer #2: Introduction should be shortened considerably

**Summary and General Comments**

Reviewer #1: I liked this manuscript. It was generally well presented and well argued. Kudos to the authors.

Reviewer #2: Onchocerciasis is one of the NTDs that is on track to elimination by 2030. However, the progress toward elimination has been uneven on the African continent. One of the countries where little is really known about transmission of O. volvulus is the Democratic Republic of Congo (DRC). The lack of knowledge concerning O. volvulus in DRC is of particular importance in the eastern areas of DRC, which border Uganda, as Uganda has made substantial progress in eliminating onchocerciasis. Thus, understanding the status of and biology of O. volvulus transmission in areas bordering Uganda are of particular urgency. This publication presents data that together represent a tour de force in the study of the entomology of O. volvulus transmission in a small focus located on the western edge of Lake Albert, bordering several foci where transmission has been eliminated in Uganda. The data support the idea that the DRC focus is isolated and represents little threat to Uganda. The study also provides support to the hypothesis that transmission in this focus is vectored by Simulium dentulosum, a species that has not been implicated as a vector until now.

I have a few comments and suggestions for improvement of the manuscript:

1. The Introduction of the manuscript is extremely long (17 pages double spaced) and reads more like an introduction to a PhD dissertation than an introduction to a research paper. I would recommend the authors edit it substantially. Reducing its length by about 2/3.

2. The authors employed informal HLCs to collect the adult flies. Normally, HLCs are given ivermectin after completing their collections, as they are at risk for exposure to O. volvulus. In this study, this risk would have been really high, given the large proportion of S. dentulosum that apparently carry infective larvae. Did the authors give the HLCs ivermectin?

3. This study provides a very detailed exploration of the flies collected but the fly numbers were quite small. 

4. The proportion of S. dentulosum with O. volvulus DNA in their heads (a marker for infectiousness) was surprisingly high – 11%. This contrasts with the areas of West Africa where hyperendemic onchocerciasis was at its worst prior to the start of the Onchocerciasis Control Program in West Africa (OCP), where the proportion of flies carrying L3 was about 5%. The authors spend a large amount of time discussing why their PCR results were not the results of a technical problem. But given the high proportion if flies likely carrying L3, confirming the PCR results by doing some simple dissections would be easy to do. Do the author have any data from dissections?

5. Given the high proportion of flies carrying L3, it seems to me that the annual transmission potential in this area would be very high, unless the flies were rather rare and the annual biting rate was correspondingly low. Can he authors calculate the annual biting rate for S. dentulosum based on their collection data, and use this to estimate the ATP?

PLOS authors have the option to publish the peer review history of their article (what does this mean?). If published, this will include your full peer review and any attached files.

Reviewer #1: No

Reviewer #2: No

Figure Files:

Data Requirements:

Reproducibility:

References

---

## [Editor Report · Decision Letter 1]

17 Oct 2022

Dear Dr Colebunders,

We are pleased to inform you that your manuscript 'Identification of the onchocerciasis vector in the Kakoi-Koda focus of the Democratic Republic of Congo' has been provisionally accepted for publication in PLOS Neglected Tropical Diseases.

Best regards,

Prof. María-Gloria Basáñez, PhD, MSc

Academic Editor

Alvaro Acosta-Serrano

Section Editor

---

## [Editor Report · Acceptance letter]

31 Oct 2022

Dear Dr Colebunders,

We are delighted to inform you that your manuscript, "Identification of the onchocerciasis vector in the Kakoi-Koda focus of the Democratic Republic of Congo," has been formally accepted for publication in PLOS Neglected Tropical Diseases.

Best regards,

Shaden Kamhawi

co-Editor-in-Chief

Paul Brindley

co-Editor-in-Chief
